# Getting the brain into gear: An online study investigating cognitive reserve and word-finding abilities in healthy ageing

**Elise J. Oosterhuis** *, **Kate Slade**, **El Smith, Patrick J. C. May, Helen E. Nuttall**

Department of Psychology, Lancaster University, Lancaster, United Kingdom

* e.oosterhuis@lancaster.ac.uk

## Abstract

Lifetime experiences and lifestyle, such as education and engaging in leisure activities, contribute to cognitive reserve (CR), which delays the onset of age-related cognitive decline. Word-finding difficulties have been identified as the most prominent cognitive problem in older age. Whether CR mitigates age-related word-finding difficulties is currently unknown. Using picture-naming and verbal fluency tasks, this online study aimed to investigate the effect of CR on word-finding ability in younger, middle-aged, and older adults. All participants were right-handed, monolingual speakers of British English. CR for both the period preceding and coinciding with the COVID-19 pandemic was measured through years of education and questionnaires concerning the frequency of engagement in cognitive, leisure, and physical activities. Linear mixed-effect models demonstrated that older adults were less accurate at action and object naming than middle-aged and younger adults. Higher CR in middle age predicted higher accuracies for action and object naming. Hence, high CR might not only be beneficial in older age, but also in middle age. This benefit will depend on multiple factors: the underlying cognitive processes, individual general cognitive processing abilities, and whether task demands are high. Moreover, younger and middle-aged adults displayed faster object naming compared to older adults. There were no differences between CR scores for the period preceding and coinciding with the pandemic. However, the effect of the COVID-19 pandemic on CR and, subsequently, on word-finding ability might only become apparent in the long term. This article discusses the implications of CR in healthy ageing as well as suggestions for conducting language production studies online.

## Introduction

The population around the world is ageing rapidly. Ageing is accompanied by cognitive decline, which starts in early adulthood [1]. Positive lifestyle choices and lifetime experiences, such as engaging in leisure activities and higher education, might delay the onset of cognitive decline and slow down its progression in healthy ageing, and could decrease the risk of dementia [2–4]. According to Cognitive Reserve (CR) theory, positive lifestyle choices and lifetime experiences build up a "cognitive reserve", which enables individuals to maintain cognitive

**Data Availability Statement:** The minimal anonymised datasets have been uploaded to the PURE repository and have the following DOIs: - Picture naming actions: https://doi.org/10.17635/

lancaster/researchdata/586 - Picture naming
objects: https://doi.org/10.17635/lancaster/
researchdata/587 - Verbal fluency actions: https://
doi.org/10.17635/lancaster/researchdata/583 -
Verbal fluency semantic: https://doi.org/10.17635/
lancaster/researchdata/588 - Verbal fluency letter:
https://doi.org/10.17635/lancaster/researchdata/
585 The study's GitHub repository (available on
https://github.com/EliseJis/GBG-Online-Study)
contains R Markdown files with the analysis code
and analysis output.

**Funding:** H.E.N. received a grant from the
Biotechnology and Biological Science Research
Council (grant number BB/S008527/1). https://
www.ukri.org/councils/bbsrc/. The funders had no
role in study design, data collection and analysis,
decision to publish, or preparation of the
manuscript. E.J.O. received a Faculty of Science &
Technology PhD scholarship from Lancaster
University. https://www.lancaster.ac.uk/sci-tech/.
The funders had no role in study design, data
collection and analysis, decision to publish, or
preparation of the manuscript.

**Competing interests:** The authors have declared
that no competing interests exist.

performance despite age-related brain changes or pathology. Greater CR can enable the use of
cognitive strategies (e.g., mnemonics), strengthen existing brain networks, and facilitate the
recruitment of additional brain networks to support cognitive performance [5, 6]. Brain
reserve is another concept often discussed in conjunction with CR. Brain reserve reflects the
brain's anatomical resources, such as number of neurons, and individuals with greater 'brain
reserve' (i.e., more neurons) are thought to be better able to tolerate neurological attrition
before it becomes pathophysiological [6]. Indeed, through 'brain maintenance' some individu-
als maintain their brains more so than others, for example, through life experience and genet-
ics. Such brain maintenance is thought to increase brain reserve [6]. Brain reserve differs from
CR in that CR explains individual differences in the ability to cope with cognitive difficulties
through differences in lifestyle choices and life experiences.

Of the cognitive difficulties older adults experience, word-finding difficulties are considered
as the most prominent problem associated with ageing [7]. Word-finding difficulties can
extend beyond the difficulties associated with healthy ageing, which could signal the onset of
dementia [8, 9]. However, CR can delay the onset of age- and dementia-related cognitive
decline [6]. The effect of CR on age-related word-finding difficulties is currently unknown.
Because word-finding difficulties have also been associated with both healthy ageing and
dementia, it would be important to investigate the link between CR and age-related word-find-
ing difficulties. This would also help us understand why some adults develop dementia whilst
others stay cognitively healthy [3, 10]. Therefore, this study aimed to investigate age-related
declines in word-finding abilities and how lifestyle factors might affect this decline. The results
of this study will inform us whether CR has a mitigating effect on age-related word-finding
difficulties.

Word-finding difficulties can be detected through picture-naming and verbal fluency tasks
[11, 12]. In picture-naming tasks, action naming has been identified as more difficult than
object naming in younger adults, possibly because different cognitive processes underly object
and action naming. More cognitive resources are needed for action naming, resulting in
slower processing times for action compared to object naming [13]. Compared to younger
adults, older adults have more difficulties in picture naming [14–18]. Age-related declines are
also found in verbal fluency [19–21], with an earlier onset and greater decline in semantic flu-
ency than in letter fluency [19, 20]. Moreover, older adults generate more common words (i.e.,
high-frequency words) in verbal fluency tasks than younger adults, which can reflect difficul-
ties in lexical access whereby highly frequent words are the easiest to access from memory
[19]. Dementia can add to the difficulties with verbal fluency tasks that are already associated
with healthy ageing [8, 9]. Therefore, verbal fluency performance could serve as an early
neuropsychological marker of dementia [22, 23].

Variability in cognitive performance across individuals increases with age (e.g. [18, 24].
Currently, there is no consensus on the age when picture-naming ability starts to decline.
Some studies indicate that the start of decline begins as early as at 30 [14] or 50 years of age
[17, 18, 25]. Yet others found that decline starts at 75 years of age, possibly because an increase
in vocabulary with age masks early word-finding difficulties [15]. The different ages of onset
might be explained by individual differences in cognitive ageing trajectories.

Differences in CR across individuals might account for the individual differences in ageing
trajectories as CR might delay the onset and decrease the rate of age-related cognitive decline
[2–4]. CR, measured through the proxy of the number of years of education, has a positive
effect on verbal and source memory [26, 27] and might modulate the effect of age on letter flu-
ency [28]. In addition, CR might not only benefit cognitive performance at older age but also
in middle-aged adults [29]. Therefore, high CR might ameliorate age-related word-finding dif-
ficulties in both middle and older age. To reduce the effect of age on word finding, CR might

strengthen the connections between brain areas involved in language processing (e.g., between the left frontal and temporal gyri) and increase bilateral activity between frontal brain regions, which enables the use of more general cognitive processes (e.g., executive functioning) to support word finding. One possibility is that participating in leisure activities and higher educational and occupational attainment train our word-finding abilities (e.g., through social interactions) and increases the efficiency of brain areas underlying word-finding [30]. However, the effect of CR on word-finding abilities across younger, middle, and older age has not yet been investigated.

The current study aimed to investigate the relationship between word-finding ability and levels of CR in young, middle-aged, and older adult participants. We hypothesised that cognitive reserve modulates age-related declines in lexical access so that higher CR predicts better performance in picture-naming and verbal fluency tasks. Furthermore, in testing this hypothesis, we first sought to replicate the results of previous studies showing that older adults have difficulties with word-finding compared to middle-aged and younger adults. Specifically, we expected an age-related decline in accuracy and reaction time for both object and action naming [16], and a decrease in the number of words produced on a verbal fluency task [18, 20]. Secondly, we expected that older adults produce more high-frequency words on verbal fluency tasks compared to younger adults [19]. In addition, because social isolation during the COVID-19 pandemic can negatively affect cognition even over a short period of time [31], we also explored whether a decrease in social and leisure activities due to the COVID-19 pandemic had an effect on CR scores and the performance of the behavioural tasks.

## Material and methods

### Participants

A total of 90 healthy, right-handed, monolingual speakers of British English participated in this study. To control for variance in word-finding speed between monolinguals and bi-/multi-linguals [32], only monolingual speakers of British English were included in this study. The sample size was based on a-priori power analysis using an online sample size calculator (https://clincalc.com/stats/samplesize.aspx; [33, 34]). The anticipated means, equating to approximately medium to large effect sizes, were based on previous literature on word-finding ability in healthy ageing [20, 21, 35–37]. The outcome variable was set as *continuous (means)*. Power was set at 80% and the alpha level at .05 to obtain a large effect size (Cohen's $f$ = 0.40). This resulted in a sample size of 30 participants per age group. The participants were grouped into 30 younger (aged 18–30; 23 females), 30 middle-aged (aged 40–55; 22 females), and 30 older adults (aged 65–80; 17 females). The age bands were based on previous literature [e.g., 21, 25, 38, 39] and the 10-year age gap between the groups was introduced to increase the sensitivity to detect age-related effects.

Participants reported no history of or current neurological (e.g., stroke or epilepsy), psychiatric (e.g., schizophrenia or bipolar), speech, or language disorders. All participants had self-reported normal or corrected-to-normal hearing and vision. All participants had a score of 19 or less on the Beck Depression Inventory-II (BDI-II; [40]) and, hence, did not suffer from depression. In addition, all participants had a score of less than 3.6 on the Informant Questionnaire on Cognitive Decline Self-Report (IQCODE-SR; [41]), reflecting absence of cognitive difficulties. Years of education, occupational score, and frequency of leisure activity can be found in Table 1. The study was approved by the Research Ethics Committee of the Faculty of Science and Technology of Lancaster University. Participants gave digital consent to take part in the study.

**Table 1. Descriptive statistics of the measures to compute cognitive reserve for the periods preceding and coinciding with the COVID-19 pandemic separately.**

| Age Group | Education (in years) | Occupational Attainment[a] | General Activities[b] | | Cognitive Activities[b] | | Social Activities[b] | | Productive Activities[b] | | Physical Activities[b] | |
|---|---|---|---|---|---|---|---|---|---|---|---|---|
| | | | Pre | During | Pre | During | Pre | During | Pre | During | Pre | During |
| Younger | 15.1 (1.6) | 9.5 (3.5) | 25.3 (8.3) | 21 (6.9) | 8.5 (3.5) | 7.7 (3.8) | 10.4 (2.9) | 5.7 (2.9) | 6.4 (3.6) | 7.6 (2.9) | 3679.4 (3988.4) | 2163 (1998.8) |
| Middle-Aged | 17 (3.1) | 4.9 (2.7) | 27.5 (6.8) | 23.1 (6.4) | 8.9 (3.1) | 8.9 (3.6) | 10 (2.5) | 4.5 (2.2) | 8.6 (3) | 9.7 (3.3) | 2418 (2068.1) | 2446.9 (1884.6) |
| Older | 18.4 (4.3) | 4.3 (1.9) | 33.7 (6.9) | 26.2 (6.8) | 13.2 (4.4) | 11.8 (4.1) | 9.9 (2.5) | 3.7 (2.6) | 10.6 (2.6) | 10.7 (3) | 2872 (1849.7) | 2149.2 (1268.3) |

[a]Lower scores on occupational attainment reflect higher socioeconomic status. NAs were present for younger (15), middle-aged (3), and older adults (9). The higher number of missing values for younger adults is due to their student status. Some older adults reported retirement status and, hence, their occupational attainment could not be calculated.

[b]Higher scores on the activities represent higher frequency of engagement in these activities.

## Materials

The BDI-II, the short IQCODE-SR, a demographics questionnaire, and the CR questionnaire were completed online via the Qualtrics XM Platform (Qualtrics, Provo, UT). The behavioural tasks for assessing word-finding ability (i.e., picture naming and verbal fluency) and general cognitive functions (i.e., cognitive processing speed, working memory, and inhibitory control) were completed online via the Gorilla Experiment Builder [42].

## Cognitive reserve

CR was quantified through a General Activities Questionnaire comprising questions about social/leisure, intellectually stimulating, and productive activities. The questions were adopted from previously published studies [43, 44] and adapted where necessary by merging, adding, or modifying questionnaire items. The frequency of participation was rated on a 4-point scale from never (0) to every day (3). In addition, physical activity was measured through the Global Activity Questionnaire [45]. The CR questionnaire was split into two parts. The first part required participants to report the frequency of their activities for the period preceding the COVID-19 pandemic. That is, the participants were asked to report the frequency of their activities retrospectively. The second part of the CR questionnaire required participants to report the frequency of their activities for the period coinciding with the COVID-19 pandemic. Occupational attainment was obtained by computing the occupational scores using the UK Standard Occupational Classification system [46]. Internal consistency was calculated using Cronbach's Alpha [47] and was $\alpha = .59$ for all items used to compute CR. Years of education, occupational attainment, and scores for the General Activities Questionnaire and Global Activity Questionnaire were converted into $z$-scores within each age group. Following the method by Soldan and colleagues (2013), a fixed composite score for CR per individual was obtained by averaging over these standardised scores [48], for the periods preceding and coinciding with the COVID-19 pandemic separately. By averaging over the $z$-scores, education, occupational attainment, and general and physical activities were given equal weight when computing CR. This resulted in two CR composite scores per individual, one for the period preceding the COVID-19 pandemic and one for the period coinciding with the pandemic.

## Picture naming

Pictures were obtained from the Center for Research in Language International Picture-Naming Project [49], and were carefully selected and balanced for word frequency (actions:

*M* = 3.3, *SD* = 1.8; objects: *M* = 3.1, *SD* = 1.7), age of acquisition of the words (actions: *M* = 5.2, *SD* = 1.7; objects: *M* = 4.8, *SD* = 1.4), and had high picture-name agreement across individuals (actions: 91%, objects: 99%). Action and object words were not matched for word length. However, this should not affect our results as word length is unlikely to influence naming speed [50]. In addition, we did not directly compare naming speed between object and action naming. The final set contained 79 pictures of actions and 70 pictures of objects. For action naming, the items "vacuum", "somersault", "wait", and "talk" were excluded as these items resulted in low accuracy or name agreement in this study. The item "cut" was excluded as this item was also used in a practice trial. For object naming, the items "football" and "skunk" were excluded as these items resulted in low name agreement across participants.

The picture naming tasks were presented in a blocked manner. That is, participants first completed the object naming task after which they completed the action naming task. Both the object and action picture-naming task started with four practice trials. The participant read the instructions on the computer screen and was instructed to name the pictures as quickly and accurately as possible. The participant first tested their microphone. Next, the participant completed four practice trials, which were recorded and played back to the participant. After the practice trials, the actual experiment started. All trials started with a fixation cross with a 500 ms duration, followed by a blank screen for 700 ms. After this, the picture was presented for 3000 ms. Differences in microphones used by the participants and the lags in audio recordings caused timing variability between participants and trials. Therefore, in order to indicate the start of the picture and measure timing variability, a trigger sound was added. The trigger sound enabled the experimenter to calculate the verbal reaction time more accurately. The stimuli were never repeated to avoid practice effects and were presented in a pseudo-random order, with the trials being randomised for frequency, age of acquisition, number of syllables in the picture names, and name agreement. Picture names starting with the same letter never followed each other. Both accuracy and reaction time were obtained using CheckVocal [51].

## Verbal fluency

The verbal fluency task comprised tasks measuring semantic, action, and letter fluency. Participants completed the letter fluency task after the semantic and action fluency task so as to avoid the use of any cued strategies. For semantic fluency, the category prompts were: "animals", "vehicles", "fruits and vegetables", "fluids", and "writing utensils". The action fluency task was prompted by "things people can do" and "things you can do to an egg". Lastly, the letter fluency task included the letter prompts "S", "M", and "P". The verbal fluency prompts were based on previous studies, which also have validated the use of these prompts [20, 21, 52, 53]. The participant had 60 seconds to produce aloud as many words as possible for each of the verbal fluency prompts. The responses were recorded and later transcribed by two transcribers in order to assess the interrater reliability. Composite scores for each of the three verbal fluency tasks were created to reduce the number of multiple comparisons. To obtain a composite score for the semantic, action, and letter fluency tasks separately, scores were standardised within age groups per verbal fluency prompt. The standardised scores were then averaged for each of the three verbal fluency tasks. This resulted in three different verbal fluency composite scores: one for semantic fluency, one for action fluency, and one for letter fluency.

## Control tasks

In order to control for the influences of more general processing difficulties in ageing [54, 55], tasks for inhibitory control, working memory, and cognitive processing speed were assessed. The control tasks were counterbalanced and presented visually and, thus, did not require

auditory processing. For all three tasks, scores were transformed into *z*-scores within age groups to create an average score per participant which reflects general cognitive processing ability. To investigate inhibitory control, a spatial Stroop task was used [56]. In this task, an arrow appeared on either the left or right side of the screen and the participant had to press the button corresponding to the direction the arrow was pointing. In incongruent trials (e.g., arrow pointing to the left but appearing on the right side of the screen), the participant had to inhibit the tendency to respond to the location of the arrow to select the correct response (i.e., the direction the arrow pointed to).

Working memory was assessed using a computerised visual adaptation of the Digit Ordering Test [56–58]. The participant was presented visually with a sequence of digits, ranging from 4 to 7 digits. Each digit in the sequence was presented for 700 ms with a 300 ms blank screen between the digits. After the sequence was presented, the participant had to type the sequence of digits in ascending order.

Finally, cognitive processing speed was assessed using the Deary-Liewald task [59], which included both Simple and Choice Reaction Time tasks. In the Simple Reaction Time task, the participant had to press the spacebar each time an X appeared in a square in the middle of the screen. In the Choice Reaction Time task, four empty squares were presented next to each other, and the X could appear in one of the four squares. The participant had to press the key corresponding to the square in which the stimulus appeared. A composite score for cognitive processing speed was created by averaging over the standardised scores of the Simple Reaction Time and Choice Reaction Time task [59]. Computerised versions of these three control tasks are widely used and even implemented in and validated for cognitive testing batteries such as the Cambridge Neuropsychological Test Automated Battery (CANTAB; [60]).

## Procedure

The participant was first assessed for their eligibility via email, after which they logged onto the Gorilla Experiment Builder. After providing digital consent, the participant was directed to Qualtrics XM Platform to complete the demographics and CR questionnaires, which included the General Activities Questionnaire and the Global Activity Questionnaire. Next, the participant was redirected to the Gorilla Experiment builder to complete the behavioural tasks. The participant started with either the three control tasks or with the language production tasks, in a counterbalanced order. For the language production tasks, the participant started first with the verbal fluency task, after which they completed the picture-naming tasks. This order was fixed to avoid priming effects of picture naming on verbal fluency performance. At the end of the experiment, the participant was debriefed and thanked for their participation. Each participant completed the CR questionnaires and the behavioural tasks only once.

## Statistical analysis

Data pre-processing and data analysis were done in R [61]. For data pre-processing, we used the package *tidyverse* [62]. For the analysis of the picture-naming data, linear mixed-effect models (LMMs) were employed, using the *lme4* package [63]. The outcome variables were 1) reaction time of correct trials for object and action naming separately and 2) accuracy for object and action naming separately. As accuracy is a binomial variable, we used generalised linear mixed-models (GLMMs) as these are well-suited for binomial data [64]. Moreover, because the accuracy data in this study is binary (1 for correct or 0 for incorrect) with a highly skewed distribution (asymmetric distribution where there are more 1's than 0's), a binomial family with a complementary log-log (cloglog) link function was chosen [65] with a "Nelder Mead" optimiser [66]. Multiple linear regression was conducted on the verbal fluency data as

there were no repeated measures. The outcome variables were 1) number of correctly produced words and 2) average frequency of the correctly produced words.

Following the preregistration, the data was trimmed using *z*-scores to reduce the probability of Type-II errors [67]. The dependent variables were transformed into *z*-scores within each age-group and outliers were identified as being above and below 2.5 *SD*. That is, picture-naming reaction times above or below 2.5 *SD* were considered outliers and subsequently removed (object naming: 4.4%, action naming: 7.9%). For verbal fluency, values for the number of correctly produced words and average frequency of the produced words above or below 2.5 *SD* were considered outliers. These were removed from the data before calculating the number of correctly produced words (semantic: 2.6%, letter: 2.9%, action: 1.7%) and average frequency of the produced words (semantic: 2.2%, letter: 2.6%, action: 2.2%). CR composite scores and scores on each of the three control measures were considered outliers if they had a value of 2.5 *SD* below or above the mean of each age group. Because there were no repeated measures for these scores and because they were important predictors in our analysis, the detected outliers were winsorised at +/- 2.5 *SD*. This allowed for the scores to remain on the extreme side of the distribution without loss of data. CR scores were missing for two participants and part of the CR scores was missing for three participants. For the control tasks, two participants had missing data for cognitive processing speed and three had missing data for working memory. Missing data was replaced using single imputation, where the mean per age group was imputed. This was done before creating any composite scores.

The predictor variables for all models were Age with three levels (younger adults aged 18–30; middle-aged adults aged 40–55; and older adults aged 65–80), the standardised CR composite score (continuous predictor), and an interaction between Age and CR composite score. The categorical variable Age was coded using Helmert contrasts such that the first contrast reflects the difference between middle-aged and younger adults, and the second contrast reflects the difference between older adults and the mean of the younger and middle-aged adult groups. The package *emmeans* was used to conduct post-hoc comparisons between age groups, including the Tukey Multiple Comparison test to obtain adjusted p-values [68]. Furthermore, we compared the models with the CR scores for the period preceding and coinciding with the COVID-19 pandemic. This was done to test whether CR scores for the period preceding or coinciding with the COVID-19 pandemic explained current word-finding ability better. Standardised scores for the general cognitive processing composite variable were added to the model as covariates.

For the picture-naming data, random effects for Item ID (i.e., variance caused by between-item variability) and Participant ID (i.e., variance caused by between-subject variability) were included [69]. To justify the inclusion of both random effects in our model, we compared the Akaike Information Criterion (AIC; [70]) values of the model with and without random effects. A lower AIC value indicates a better fit. For action naming reaction time, the AIC for inclusion of both random effects was substantially lower than the null model without random effects (32.9 and 2095.4 respectively). For object naming reaction time, the model which included both random effects resulted in a substantially lower AIC value compared to the null model without random effects (-796.3 and 1403.3 respectively). For action naming accuracy, the AIC for the model with both random effects was lower than the null model without random effects (2771.5 and 2994.4 respectively). For the model with both random effects for object naming accuracy, the AIC was also lower than the null model without random effects (1012.50 and 1070.89 respectively).

To assess goodness of fit, the assumptions of the different statistical models were investigated, and the AIC was compared against a null model. In addition, we reported the conditional and marginal R-squared ($R^2$) for all models using the function "r2_nakagawa" of the

package *performance* [71] for the LMMs and the function "r.squaredGLMM" of the package *MuMIn* for the GLMMs [72]. Moreover, goodness of fit of the GLMMs was assessed by testing for overdispersion, using "testDispersion" function of the *DHARMa* package [73]. Concordance and Somer's D were calculated using the "somers2" function of the *Hmisc* package to assess the predictive power of the GLMMs [74].

### Data availability

The study hypotheses, design, and statistical analyses were preregistered on aspredicted.org (https://aspredicted.org/b6we4.pdf). R code, Cognitive Reserve questionnaires, picture names, and additional online materials are openly available at the project's GitHub repository (https://github.com/EliseJis/GBG-Online-Study). The dataset will be openly available via Lancaster University's Pure repository. The dataset's DOI will be made available after acceptance.

## Results

### Picture naming reaction time data

The reaction times for action and object naming were analysed separately. Mean reaction times per age group were calculated for accurate items only. The mean reaction time for action naming was 1063.9 ms (*SD* = 348.3 ms) for younger adults, 1006.6 ms (*SD* = 312.4 ms) for middle-aged adults, and 1071.2 ms (*SD* = 326.8 ms) for older adults. The mean reaction time for object naming was 763.6 ms (*SD* = 183.2 ms), 792.9 ms (*SD* = 222.4 ms), and 881.0 ms (*SD* = 246.9 ms) for younger, middle-aged, and older adults, respectively.

The full model for both action and object naming with main predictors (Age and CR), the interaction term (Age*CR) and the covariates (the control tasks) converged without the need for optimisers. The assumptions for linearity and homoscedasticity were both met. To obtain normality of residuals, the outcome variable (reaction time) was log transformed.

Model formulas and results are reported in Tables 2 and 3 for action and object naming respectively. For action naming reaction time, none of the predictors were significant. For object naming reaction time, we found a main effect of age ($\beta$ = 0.05, $t(87.99)$ = 3.83), such that older adults were significantly slower than both younger (adjusted $p$ = .003) and middle-aged adults (adjusted $p$ = .001). The interaction between CR and Age did not reach significance.

For action naming, approximately 16.2% and 16.8% of the variance not explained by our fixed effects was explained by the random effects for Participant ID and Item ID respectively. Marginal and conditional $R^2$ were calculated as a measure of model fit. Fixed effects explained 1.4% of the variance in the data (marginal $R^2$ = .014), whilst 34.0% of the variance was explained by the full model, including random effects (conditional $R^2$ = .340). For object naming, 24.3% and 9.2% of the variance not explained by our fixed effects was explained by the random effects for Participant ID and Item ID, respectively. Fixed effects explained 5.9% of the variance in the data (marginal $R^2$ = .059), whilst 37.4% of the variance was explained by the full model, including random effects (conditional $R^2$ = .374).

### Picture naming accuracy data

Accuracy data for object and action naming was analysed separately. Mean accuracy for action naming was 94.7% for younger adults (*SD* = 22.4%), 95.4% for middle-aged adults (*SD* = 20.9%), and 93.5% for older adults (SD 24.7%). For object naming, the respective mean accuracies were 99% (*SD* = 10%), 98.7% (*SD* = 11.3%), and 97.3% (*SD* = 16.1%).

**Table 2. Results of the fixed effects for action naming reaction time.**

| Model | log(RT) ~ Age Group*CR score + General Cognitive Processing + (1\|Participant ID) + (1\|Item ID) | | | | | | |
|---|---|---|---|---|---|---|---|
| **Effect** | **Estimate** | **SE[d]** | **t value** | **df[e]** | **p value** | **CI[f] Low** | **CI[f] High** |
| Intercept | 6.904 | 0.018 | 381.321 | 155.999 | .000*** | 6.868 | 6.940 |
| MA[a] vs. YA[b] | 0.019 | 0.016 | 1.156 | 89.661 | .251 | -0.013 | 0.051 |
| OA[c] vs. MA[a] / YA[b] | 0.015 | 0.010 | 1.434 | 89.638 | .155 | -0.006 | 0.035 |
| CR[g] Score | -0.006 | 0.013 | -0.436 | 89.726 | .664 | -0.031 | 0.020 |
| MA[a] vs. YA[b] * CR[g] Score | 0.012 | 0.016 | 0.742 | 89.670 | .460 | -0.019 | 0.042 |
| OA[c] vs. (MA[a] / YA[b]) * CR[g] Score | -0.014 | 0.009 | 1.555 | 89.859 | .124 | -0.032 | 0.004 |
| Covariates | | | | | | | |
| General Cognitive Processing | -0.009 | 0.029 | -0.310 | 88.582 | .757 | -0.066 | 0.048 |

[a]MA = middle-aged adults

[b]YA = younger adults

[c]OA = older adults

[d]SE = standard error

[e]df = degrees of freedom

[f]CI = 95% Confidence Intervals

[g]CR = Cognitive Reserve

*$p < .05$

**$p < .01$

***$p < .001$.

Model formulas and the results of both models are reported in Tables 4 and 5 for action and object naming respectively. For both models, there was a main effect for Age, in that older adults were less accurate in action and object naming than middle-aged adults (action naming:

**Table 3. Results of the fixed effects for object naming reaction time.**

| Model | log(RT) ~ Age Group*CR score + General Cognitive Processing + (1\|Participant ID) + (1\|Item ID) | | | | | | |
|---|---|---|---|---|---|---|---|
| **Effect** | **Estimate** | **SE[d]** | **t value** | **df[e]** | **p value** | **CI[f] Low** | **CI[f] High** |
| Intercept | 6.679 | 0.017 | 388.170 | 141.602 | .000 | 6.586 | 6.690 |
| MA[a] vs. YA[b] | -0.004 | 0.019 | -0.207 | 87.845 | .837 | -0.082 | 0.067 |
| OA[c] vs. MA[a] / YA[b] | 0.045 | 0.012 | 3.834 | 87.952 | < .001*** | 0.058 | 0.202 |
| CR[g] Score | -0.005 | 0.015 | -0.369 | 87.953 | .713 | -0.062 | 0.036 |
| MA[a] vs. YA[b] * CR[g] Score | 0.024 | 0.018 | 1.332 | 87.887 | .186 | -0.024 | 0.119 |
| OA[c] vs. (MA[a] / YA[b]) * CR[c] Score | -0.016 | 0.010 | -1.595 | 88.018 | .114 | -0.095 | 0.045 |
| Covariates | | | | | | | |
| General Cognitive Processing | 0.019 | 0.033 | 0.578 | 87.924 | .564 | -0.010 | 0.085 |

[a]MA = middle-aged adults

[b]YA = younger adults

[c]OA = older adults

[d]SE = standard error

[e]df = degrees of freedom

[f]CI = 95% Confidence Intervals

[g]CR = Cognitive Reserve

*$p < .05$

**$p < .01$

***$p < .001$.

**Table 4. Results of the fixed effects for action naming accuracy.**

| Model | Accuracy ~ Age Group*CR score + General Cognitive Processing + (1\|Participant ID) +(1\|Item ID), family = binomial (link = "cloglog") | | | | | |
|---|---|---|---|---|---|---|
| Effect | Estimate | SE[d] | z value | p value | CI[e] Low | CI[e] High |
| Intercept | 1.255 | 0.053 | 23.826 | .000 | 1.152 | 1.358 |
| MA[a] vs. YA[b] | -0.037 | 0.035 | -1.083 | .279 | -0.105 | 0.030 |
| OA[c] vs. MA[a] / YA[b] | -0.053 | 0.021 | -2.259 | .011* | -0.094 | -0.012 |
| CR[f] Score | 0.023 | 0.027 | 0.864 | .387 | -0.029 | 0.076 |
| MA[a] vs. YA[b] * CR[f] Score | -0.113 | 0.033 | -3.379 | < .001*** | -0.178 | -0.047 |
| OA[c] vs. (MA[a] / YA[b]) * CR[f] Score | -0.007 | 0.019 | -0.363 | .716 | -0.043 | 0.030 |
| Covariates | | | | | | |
| General Cognitive Processing | -0.021 | 0.059 | -0.360 | .718 | -0.138 | 0.095 |

[a]MA = middle-aged adults

[b]YA = younger adults

[c]OA = older adults

[d]SE = standard error

[e]CI = 95% Confidence Intervals

fCR = Cognitive Reserve

*p < .05

**p < .01

***p< .001.

$\beta$ = 0.20, $z$ = 2.92, adjusted $p$ = .010; object naming: $\beta$ = -0.19, $z$ = 2.33, adjusted $p$ = .05) and less accurate in object naming than younger adults ($\beta$ = 0.25, $z$ = 2.53, adjusted $p$ = .03). For action naming accuracy, the interaction between Age and CR reached statistical significance in

**Table 5. Results of the fixed effects for object naming accuracy.**

| Model | Accuracy ~Age Group*CR score + General Cognitive Processing + (1\|Participant ID)+ (1\|Item ID), family = binomial (link = "cloglog") | | | | | |
|---|---|---|---|---|---|---|
| Effect | Estimate | SE[d] | z value | p value | CI[e] Low | CI[e] High |
| Intercept | 1.630 | 0.069 | 23.781 | .000 | 1.499 | 1.768 |
| MA[a] vs. YA[b] | 0.031 | 0.048 | 0.653 | .514 | -0.062 | 0.125 |
| OA[c] vs. MA[a] / YA[b] | -0.074 | 0.026 | -2.846 | .004** | -0.125 | -0.023 |
| CR[g] Score | -0.013 | 0.034 | -0.385 | .701 | -0.079 | 0.053 |
| MAa vs. YA[b] * CR[f] Score | -0.106 | 0.045 | -2.368 | .018* | -0.193 | -0.018 |
| OA[c] vs. (MA[a] / YA[b]) * CRf Score | 0.011 | 0.022 | 0.503 | .615 | -0.032 | 0.054 |
| Covariates | | | | | | |
| General Cognitive Processing | 0.047 | 0.073 | 0.644 | .520 | -0.096 | 0.189 |

[a]MA = middle-aged adults

[b]YA = younger adults

[c]OA = older adults

[d]SE = standard error

[e]CI = 95% Confidence Intervals

fCR = Cognitive Reserve

*p < .05

**p < .01

***p< .001.

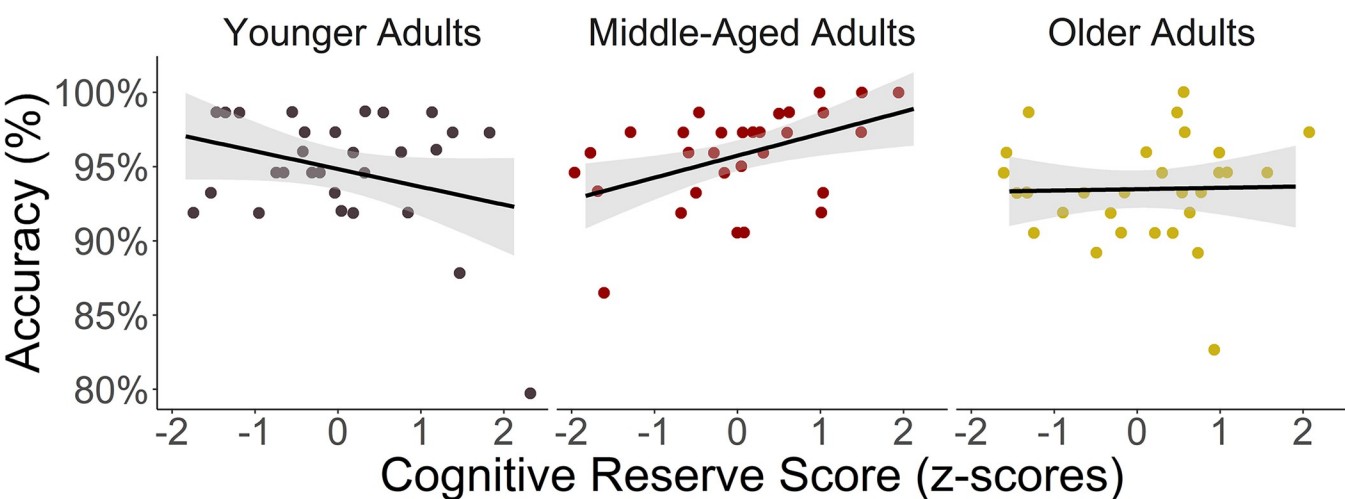

**Fig 1. The relationship between Cognitive Reserve (CR) and action naming accuracy per age group.** The Effect of CR is Significant in Middle-Aged Adults.

middle-aged adults (see Fig 1). That is, CR had a positive effect on action naming accuracy ($\beta$ = -0.11, $z$ = -3.38, $p$ < .001). Post-hoc comparisons revealed that the interaction was significant in middle-aged compared to younger adults (adjusted $p$ = .002). For object naming accuracy, we found an interaction effect between Age and CR ($\beta$ = -0.11, $z$ = -2.37, $p$ = .020). Post-hoc comparisons revealed that the interaction was significant in middle-aged compared to younger adults (adjusted $p$ = .047; see Fig 2).

Both the models for object and action naming showed a good fit for picture-naming accuracy and there was no overdispersion. For action naming, Concordance and Somer's $D$ were .82 and .65, respectively. For object naming, the respective values were .89 and .78. In addition, marginal and conditional $R^2$ for binomial distributions were calculated. Marginal $R^2$ was .008 for action naming and .012 for object naming. Conditional $R^2$ was .096 for action naming and .078 for object naming. However, logistic models often lead to low $R^2$ values [75] and low $R^2$ values are not an indication of poor model fit but could indicate a wider spread of data points instead [76, 77].

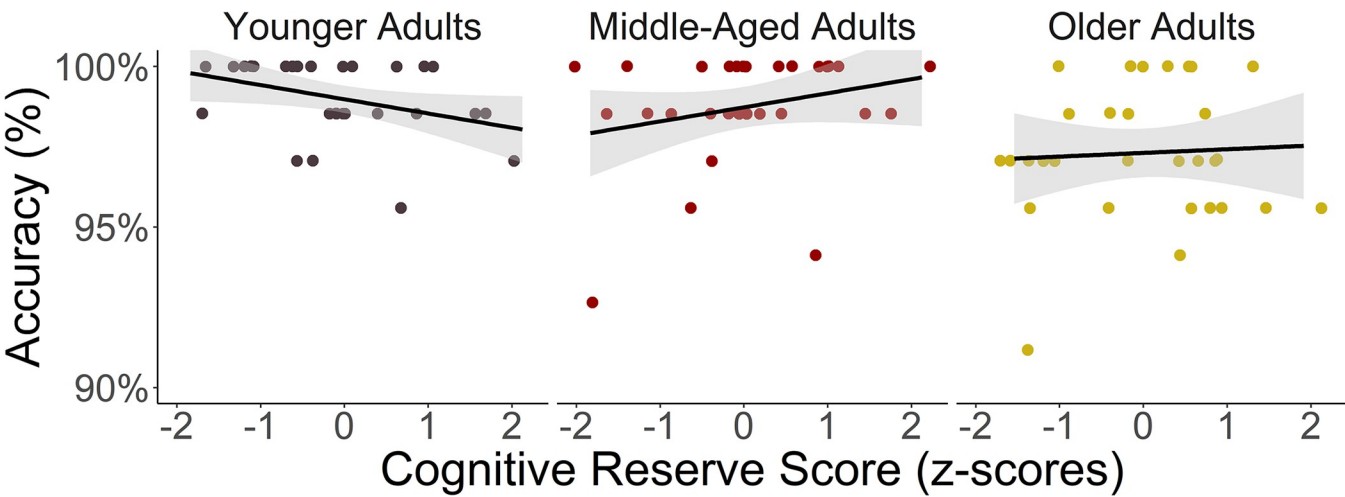

**Fig 2. The relationship between Cognitive Reserve (CR) and action naming accuracy per age group.** The Effect of CR is Significant in Middle-Aged Adults.

**Table 6. Descriptive statistics of the verbal fluency tasks (i.e., semantic, letter, and action fluency) per age group.**

| Age Group | Measure | Task[b] | | |
|---|---|---|---|---|
| | | **Semantic** | **Letter** | **Action** |
| Younger adults | N words total[a] | 88 (19) | 48 (14) | 36 (7) |
| | Average Frequency | 3.96 (0.16) | 4.22 (2.64) | 4.20 (0.17) |
| Middle-aged adults | N words total | 94 (18) | 54 (18) | 36 (9) |
| | Average Frequency | 4.40 (2.99) | 4.08 (0.26) | 4.21 (0.16) |
| Older adults | N words total | 85 (14) | 51 (20) | 33 (6) |
| | Average Frequency | 4.23 (2.05) | 4.92 (0.24) | 4.32 (0.88) |

[a]N words total is the total number of correctly produced words.

[b]Values reflect means (SD)

### Verbal fluency data

Table 6 summarises the descriptive statistics of the verbal fluency tasks. For the models with the number of correctly produced words as the outcome variable, General Cognitive Processing was a significant predictor for action fluency only ($\beta$ = 0.42, $t$(83) = 2.25, $p$ = .03). None of the other predictors were significant. This was the case for all three verbal fluency variables (i.e., semantic, action, and letter fluency). For semantic fluency, Age was a significant predictor of average frequency of the produced words ($\beta$ = -0.06, $t$(83) = 2.31), but the effect disappeared after correcting for multiple comparisons ($p$ = .11). Moreover, none of the models predicted the data well, as was assessed through model fit diagnostics, including a non-significant overall model fit.

### The effect of COVID-19 on CR and subsequent task performance

It is possible that social isolation due to the COVID-19 pandemic led to changes in CR scores, which could impact current word-finding ability. Table 1 provides the descriptive statistics per age group for the CR scores preceding and coinciding the COVID-19 pandemic. To investigate whether changes in CR scores due to the COVID-19 pandemic affected behavioural performance, we performed an exploratory analysis whereby we ran all models again with the CR scores measured during the pandemic. Next, ANOVA for model comparisons and AIC were used to compare the models with CR scores preceding the COVID-19 pandemic to that with the CR scores coinciding with the pandemic for all picture-naming and verbal fluency models separately. The ANOVAs for model comparisons and AIC values revealed no differences between the models with CR scores preceding the COVID-19 pandemic and those with CR scores coinciding the pandemic.

### Discussion

This study investigated the relationship between CR and age-related word-finding difficulties, and explored the impact of the COVID-19 pandemic on CR. To our knowledge, this is the first study investigating these relationships. We hypothesised that CR modulates age-related declines in word finding. We quantified word-finding difficulties through picture-naming and verbal fluency tasks, and CR through a comprehensive questionnaire on lifestyle. We found that age predicts word-finding ability: Older adults had lower accuracy of action and object naming compared to middle-aged adults, and they had lower accuracy of object naming compared to younger adults. Both middle-aged and younger adults displayed faster reaction times for object naming compared to older adults. We found no age effect on action naming reaction

time nor on any of the verbal fluency measures. In addition, middle-aged adults with higher CR scores reached higher accuracies for both action and object naming. CR did not modulate action or object naming reaction times, nor did it modulate verbal fluency performance in any of the age groups. Finally, CR scores preceding the COVID-19 pandemic did not explain behavioural scores any differently than CR scores coinciding with the pandemic.

We found that older adults achieved lower accuracy for action and object naming and were slower in object naming, which is in line with previous studies showing a decrease in word-finding ability in older age [25, 36]. Older age did not predict reaction times for action naming. A recent study showed that action naming was less affected by age, possibly because action and object naming rely on different neural networks [78]. In addition, research suggests that action naming is a slower process compared to object naming because more cognitive resources are needed [13]. Hence, the speed of accessing action words may be relatively preserved in older adults, due to already slower processes involved and because of different underlying neural networks.

Our study did not reveal an effect of general cognitive processing on picture-naming performance. Some ageing theories state that cognitive deficits are caused by a shrinking of cognitive resources, such as a slowing down of cognitive processing speed [55] or a reduction of working memory capacity [54]. If the above-mentioned theories are correct, an age-related reduction in cognitive resources would explain the variance in the reaction times for action and object naming in the current study. Note this point holds true only under the assumption that the measures employed are good measures for capturing the aforementioned constructs. However, pure measures of complex psychological constructs, such as cognitive processing speed, are difficult to capture because measures can be influenced by multiple factors in addition to the construct itself [79]. However, general cognitive processing scores did not predict word finding in the current study. Hence, the age-related word-finding difficulties during the picture-naming task could not be explained by a reduction in cognitive resources. Future research should investigate the influence of general cognitive processes in relation to CR on word finding abilities.

We found no interactions between age and CR for reaction times nor for accuracies of action and object naming in older adults, which is in contrast with studies showing a modulatory effect of CR on cognitive functioning in older age [2]. There may be two possible reasons for this discrepancy. Firstly, previous studies showed that CR does not benefit all cognitive domains equally [26, 79]. In these studies, whilst CR was associated with better verbal memory and crystallised intelligence in older adults, CR did not enhance performance on a finger tapping, cognitive processing speed, or executive functioning task. Hence, CR might not mitigate word-finding difficulties in older age. Alternatively, CR might increase performance when task demands are high by enhancing general cognitive processes (for a discussion, see [80]). Hence, the effects from CR might become too subtle to detect whilst the beneficial effects on word finding from other cognitive processes becomes apparent.

In contrast to the findings in older adults, our results revealed an interaction between CR and age for action and object naming accuracy in middle-aged adults, with a stronger effect for action than object naming. Previous studies have reported that CR, quantified through years of education, modulates the performance of verbal memory and semantic memory [26, 27]. To our knowledge, the current study is the first to demonstrate that middle-aged people with high CR achieve higher action naming accuracy than those with low CR. Action naming is more challenging than object naming and the two processes are quite different from one another because of the different underlying cognitive processes and brain regions [13]. Hence, if CR is drawn upon when task demands are high and additional resources are needed, for example due to slower cognitive processes underlying action-word finding ability [5], it would benefit action naming more than object naming.

Moreover, the modulating effect of CR in middle-aged adults is in line with the findings of Cansino and others [29], who found that CR only contributes to source memory performance in middle age but not in older age. In middle age, high CR might be necessary as an additional resource when task demands are high [5], whilst in younger and older adults, different cognitive strategies are necessary to tackle difficult tasks [29]. Therefore, middle-aged adults might draw upon CR resources to support their word-finding abilities whilst older adults might enhance their word-finding abilities by tapping into other cognitive processes, such as cognitive processing speed, working memory, or vocabulary, that might be more beneficial [29]. A cautionary note should be given as it is currently unclear whether there is a causal link between CR and cognitive functioning. Several factors can influence cognitive ageing that have not been measured here, such as hearing loss [81], smoking, and high blood pressure [82], to name a few. Such factors have not been considered in the current study but could in theory have exerted influence on the relationship between CR and word-finding abilities. Longitudinal studies that control for such factors and investigate the causal role of CR interventions on cognition would provide critical information regarding the causal link between CR and cognition.

Furthermore, we observed that older adults experienced greater word-finding difficulties relative to both younger and middle-aged adults. Currently, there is no consensus on the age when word-finding difficulties become apparent. The results of this study indicate that subtle word-finding abilities can be detected from the age of 65 years old. This is in line with previous studies suggesting that a decline in word-finding abilities starts at 75 years of age [15]. Some studies suggested that word-finding difficulties already become apparent at the age of 30 years old [14] or as early as 50 years of age [17, 18, 25]. In contrast, we did not find differences in word-finding abilities between younger and middle-aged adults. Hence, it is likely that subtle word-finding difficulties can commence from the age of 65, depending on the individual.

The COVID-19 pandemic might have led to changes in CR scores, which could have immediately impacted word-finding ability. The current study investigated whether the level of CR coinciding with the COVID-19 pandemic would explain the data better than the CR levels preceding the pandemic. The current study showed that CR scores for the period coinciding with the COVID-19 pandemic did not explain behavioural performance better than CR scores for the period preceding the pandemic. Assuming that the pandemic resulted in short-term changes in CR levels, these changes do not seem to influence behaviour. One explanation is that there is a "critical period" for accumulating CR [83]. Therefore, CR accumulated early in life might be more important for word-finding ability than CR accrued later. As a result, changes in CR in older age would not modulate age-related declines in word-finding ability. However, several studies demonstrate that increasing CR levels at older age, for example through late-life education, increases cognitive performance [4, 79]. Moreover, the current study measured CR in both early (i.e., education) and later life (i.e., current engagement in leisure, cognitive, and psychical activities), and showed that CR modulated performance in middle and older age. Alternatively, the effects of lower CR on word-finding ability might be delayed and only become apparent later on. If reductions in CR scores result in lower cognitive performance in the long term, this could have an enormous impact on healthy brain function later in life. Therefore, it is of paramount importance that future studies investigate the long-term effects of changes in CR due to the COVID-19 pandemic on cognition.

Regarding verbal fluency, we found no effect of age or CR. In contrast, previous studies reported age-related decreases in verbal fluency performance [19–21] and a modulatory effect of CR on letter fluency [28]. The study was conducted online (because of the COVID-19 pandemic) and as such, participants had no opportunity to ask the researcher directly for clarification of the instructions. Because verbal fluency tasks are normally conducted in-person,

administering this task via an online platform without the researcher being present might have resulted in null results for verbal fluency in this study.

The optimal way to quantify CR is currently unclear as there are several proxies that can contribute to CR [6]. However, using a single proxy for CR, such as years of education, might not fully capture CR. In order to obtain a score that reflects CR across the lifespan, the current study implemented a more comprehensive measure by creating a composite score from the frequency of leisure, social, and cognitively stimulating activities in addition to educational and occupational attainment.

Lastly, we found that quite a large proportion of the variance in the picture naming reaction time data was explained by the additional random effects for between-individual and between-trial variability (when comparing conditional vs. marginal $R^2$). Previous studies showed that ageing trajectories are very person-specific and the deviation in performance increases with age [18, 24]. Hence, the large between-individual variability highlights the importance of controlling for individual differences in ageing research.

## Limitations

The online nature of the study meant that the participants were not in a controlled environment. We tried to control the study through initial screening, providing clear instructions with example pictures, giving the opportunity to test the microphone, and by asking the participants to complete the study in a quiet environment with as few distractions as possible. Future online studies would benefit from a researcher attending the session via a conference call whilst the participant is completing the study to increase participant-researcher interaction. Furthermore, the current study was restricted to a sample of monolingual speakers of British English to reduce the variance in word-finding speed, which can be affected by language background [32]. Hence, the results of this study cannot be generalised to multilingual speakers. Future studies should investigate this question using groups of speakers with varied language backgrounds to expand on the relationships between CR and word-finding abilities across the lifespan. Such future research will increase our understanding of the effects observed in the current study, which has a restricted sample.

In the current study, we transformed the picture-naming reaction times and the number of correctly produced words on the verbal fluency tasks into $z$-scores within each participant group. We then classified any score above or below 2.5 $SD$ as an outlier, which was subsequently removed. It is theoretically possible that trimming data, including the above-mentioned method, can lead to distortions of the data set [84, 85]. However, recent research suggests that not excluding outliers leads to a greater negative bias [67]. That is, the probability of Type-II errors increases when the data is not trimmed. Using the $z$-score method mentioned in this study leads to the smallest bias. With increasing age, standard deviations in cognitive tasks increase, meaning more variance within older than younger samples. Hence, trimming the data, especially in older adults, may lead to masking of deviation effects. In the current study, we chose to trim the data within each of the age groups, reducing the chance of masking such deviations. In addition, it is likely that trimming the data using $z$-scores has led to lower bias than not trimming the data at all (see [83]).

We also note that the study was powered for a medium-large effect size, so any smaller effects sizes may have been missed. Future work could use the data collected here to inform data simulations or power calculations associated with specific statistical tests. For LMMs, Hox proposes to use the rule of thumb of 30 groups with 30 observations [86]. The current study has 30 participants per group and over 70 trials per participants, and, hence, would comply with Hox's suggestion, so it is likely that the study is adequately powered.

The internal consistency of the CR questionnaire was poor ($\alpha$ = .59), indicating that the items in the questionnaire are multidimensional [87]. The low internal consistency score could be due to few questionnaire items or because they do not measure the same underlying construct. CR is a concept that is believed to be built up by lifestyle choices and lifetime experiences, such as physical activity and education [6]. Hence, CR in itself is a multidimensional construct and this multidimensionality might be reflected through the low Cronbach's Alpha score. However, there is currently no consensus as to what lifestyle choices and lifetime experiences must be included to compute an individual's CR level (for a discussion, please see [80]). To better capture CR, the current study included a range of CR proxies, including years of education, frequency of cognitively stimulating activities, and physical activities. Hence, the low Cronbach Alpha score may be a trade-off of using a more comprehensive CR measure.

Moreover, the use of different operating systems, browsers, and differences in microphone setup could have increased the variability in scores between participants. To reduce the timing variability in the audio recordings for the picture-naming tasks, the start of each trial was indicated with a trigger sound. To increase the reliability of the estimate of reaction time, we subtracted the trigger onset from voice onset. However, we could not control for the distance between participant and microphone, and this could have increased reaction time variability across participants and even across trials. We recommend that future online studies provide instructions on the distance between the participant and the computer (e.g., the participant should sit at arm's length).

Lastly, the current study did not investigate the effect of vocabulary on picture-naming performance in older adults. Previous studies showed that vocabulary positively modulated word-finding ability in older adults [29] and that an increase in vocabulary masks subtle declines in word-finding ability [38, 88]. However, we found age-related declines in word-finding ability in older adults. This participant group was slower in object naming and achieved lower accuracy for action and object naming than younger and middle-aged adults. Because word-finding difficulties in older age were present in the current study, vocabulary could not have masked word-finding difficulties. However, future studies on CR need to clarify the effect of both vocabulary and CR on word finding across the lifespan to provide a clearer picture as to what resources are necessary in maintaining word-finding in older age.

## Conclusions

Lower action and object naming accuracy, and slower object naming reaction times, are associated with older age. These results suggest the presence of subtle word-finding difficulties in older age. Moreover, the results showed that high CR is associated with more accurate action and object naming in middle age. Hence, high CR might not always be beneficial in older age, but also in middle age, and this benefit will depend on multiple factors: the underlying cognitive processes, whether task demands are high, and individual differences in general cognitive processing abilities. Future studies should clarify at which life stages CR is most beneficial and whether this depends on the cognitive function being investigated. Finally, changes in CR due to the COVID-19 pandemic did not affect word-finding abilities. However, such effects might become apparent only after several years, and future studies should investigate the long-term effects of the COVID-19 pandemic on CR and subsequent cognitive functioning. If changes in CR negatively affects cognitive functioning, timely interventions are necessary to counteract the negative consequences and increased risk of dementia by, for example, increasing late-life CR. Such interventions can include cognitive training, education later in life, and promoting healthy lifestyle choices throughout the lifespan (for a discussion on interventions, please see [80]).

## Acknowledgments

We would like to thank the members of the Neuroscience of Speech and Action laboratory for their support. We thank Peter Tovee and Barrie Usherwood for their advice on setting up a language production study using online platforms. Lastly, we would like to thank all the participants for their time and efforts.

## Author Contributions

**Conceptualization:** Elise J. Oosterhuis, Helen E. Nuttall.

**Data curation:** Elise J. Oosterhuis.

**Formal analysis:** Elise J. Oosterhuis, Kate Slade, Patrick J. C. May, Helen E. Nuttall.

**Investigation:** Elise J. Oosterhuis, El Smith.

**Methodology:** Elise J. Oosterhuis, Kate Slade, Patrick J. C. May, Helen E. Nuttall.

**Project administration:** Elise J. Oosterhuis, Kate Slade, Patrick J. C. May, Helen E. Nuttall.

**Resources:** Elise J. Oosterhuis.

**Supervision:** Kate Slade, Patrick J. C. May, Helen E. Nuttall.

**Validation:** Elise J. Oosterhuis.

**Visualization:** Elise J. Oosterhuis, Helen E. Nuttall.

**Writing – original draft:** Elise J. Oosterhuis.

**Writing – review & editing:** Elise J. Oosterhuis, Kate Slade, Patrick J. C. May, Helen E. Nuttall.

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
