## [Decision Letter · Decision Letter 0]

4 Nov 2022

PONE-D-22-23466Getting the brain into gear. An online study investigating cognitive reserve and word-finding abilities in healthy ageingPLOS ONE

Dear Dr. Oosterhuis,

Thank you for submitting your manuscript to PLOS ONE. After careful consideration, we feel that it has merit but does not fully meet PLOS ONE’s publication criteria as it currently stands. Therefore, we invite you to submit a revised version of the manuscript that addresses the points raised during the review process.

We look forward to receiving your revised manuscript.

Kind regards,

Katya Numbers, PhD, M.S., B.S.

Academic Editor

PLOS ONE

Journal Requirements:

Additional Editor Comments:

Though we noted some strengths to the manuscript, such as a focus on the important topic of cognitive reserve, there were also some critical concerns that came up in the reviews and in my own reading. Please address the reviewers' comments below and feel free to be in touch should you have any questions. We look forward to an updated draft soon. 

Reviewers' comments:

Reviewer's Responses to Questions

**Comments to the Author**

1. Is the manuscript technically sound, and do the data support the conclusions?

Reviewer #1: Yes

Reviewer #2: Partly

2. Has the statistical analysis been performed appropriately and rigorously? 

Reviewer #1: Yes

Reviewer #2: Yes

3. Have the authors made all data underlying the findings in their manuscript fully available?

Reviewer #1: No

Reviewer #2: No

4. Is the manuscript presented in an intelligible fashion and written in standard English?

Reviewer #1: Yes

Reviewer #2: Yes

5. Review Comments to the Author

Reviewer #1: This is a well designed and mostly clearly written study on an important topic. The (important) pandemic aspect of the data is however not explained enough to the extent that the reader could completely follow.

Major

- The truncation of the data seems odd to me (and not without consequences, see e.g., Jeff Miller's work), especially in the context of studying populations that are likely to start deviating more (i.e., the older group), a deviation that is relevant. But even if the authors choose to trim the data, it would be important to be explicit about how the 2.5 SD was calculated: over participants, within participants, per task, etc? This is made explicit in line 263 but it would be important to clarify. In any case, especially in the older group, more extreme values could be relevant for the research question rather than outliers. If the authors choose to keep the procedure of trimming the data, I would suggest they do mention the potential issues.

- lines 429-435: The discussion on cognitive resources in ageing is put a bit simplistically in my view. The statement "If these theories are correct [...]" (line 429) is true critically under the assumption that the measures employed are good measures for capturing those constructs. Unfortunately, the state of cognitive sciences/neuropsychology is such that this assumption barely ever holds in its strong form. I would consider making this hidden premise explicit in the discussion.

- Related to the above: a cautionary note could be given on the underlying assumption of a causal link between CR  cognitive functions. This remains a tricky issue: Can one really know for sure that it's CR that affects cognitive functioning, rather than any other causal relationship (that one did not measure)?

- page 7, lines 160-163: This is unclear, at least at this point, the reader has no way of knowing if participants were tested twice, once pre and once during, or why in any case there would be one measurement preceding and one coinciding with the pandemic. Moreover, this issue is not clarified at any point in the manuscript.

Minor (in no particular order)

- lines 253-254: "Multiple linear regression was conducted on the verbal fluency data as there were no repeated measures". In a way, there are since there were multiple prompts per fluency task. Is there a reason why the authors chose to work with averaged scores for this task (cf. picture naming)?

- were objects and actions also matched for word length? Although not a critical variable (e.g., Alario et al. 2004), it could be helpful for the reader to provide that information

- lines 259-262: I do not understand why the percentage of removed data differs depending on whether one is looking at number of produced words or their frequency. Shouldn't they be the same, i.e., data is removed and from that one calculates the number of produced words and their frequency?

- lines 283-284: "random effects for Trial (i.e., variance caused by between-item variability)", why trial and not item ID? In my understanding, the item ID will capture the between-item variability, trial will not; instead, it will capture variability due to passing of time (given that items were presented in randomised fashion).

- 2.2.2: Please, clarify if objects and actions were presented in a blocked manner or fully randomised. Given the sentence in line 175, "Both the object and action picture-naming task", I assume they were blocked but it would be helpful to have this information explicit.

- page 3, "Word-finding difficulties can extend beyond the difficulties associated with healthy ageing in dementia (8,9)." Is there a phrasing issue here "healthy ageing in dementia"?

- page 7, lines 167-168: are the number in parentheses SD? Please, indicate.

- lines 247-248: "reaction time of correct trials only and 2) accuracy for object and action naming separately": This suggests the RT data was collapsed for naming task whereas the accuracy data was not. I don't think that was the case but maybe the wording can be made clearer?

- First paragraph of Limitations: I do not see how this is a limitation of the study. Rather, I would consider it a strength that the authors went further than just using years of education as a measure for CR.

References

Alario, F.-X., Ferrand, L., Laganaro, M., New, B., Frauenfelder, U. H., & Segui, J. (2004). Predictors of picture naming speed. Behavior Research Methods, Instruments, & Computers, 36(1), 140–155. https://doi.org/10.3758/BF03195559

Miller, J. (1991). Reaction Time Analysis with Outlier Exclusion: Bias Varies with Sample Size. The Quarterly Journal of Experimental Psychology Section A, 43(4), 907–912. https://doi.org/10.1080/14640749108400962

Ulrich, R., & Miller, J. (1994). Effects of truncation on reaction time analysis. Journal of Experimental Psychology: General, 123(1), 34–80. https://doi.org/10.1037/0096-3445.123.1.34

Reviewer #2: Peer review report on “Getting the brain into gear. An online study investigating cognitive reserve and wordfinding abilities in healthy ageing”

1. Original submission

1.1. Recommendation

Major Revision

2. Comments to Author:

Ms. Ref. No: PONE-D-22-23466

Title: Getting the brain into gear. An online study investigating cognitive reserve and wordfinding abilities in healthy ageing

Authors: Oosterhuise, E.J., Slade, K., Smith, E., May, P.J.C., Nuttal, H.E.

Overview and general recommendation:

Cognitive reserve (CR) is a potentially modifiable risk factor for cognitive decline and dementia. The authors examined CR in younger, middle-aged and older adults and its association with picture-naming and verbal fluency tasks. The authors ran an online study before and during COVID, and ran linear mixed effect models to analyse the data. They found that younger and middle-aged adults were faster at object naming compared to older adults. They also found that in middle age, CR predicted higher accuracy in action and object naming, suggesting that CR might be important in middle-age.

The study has several strengths. There is a lack of life-span approaches in the healthy ageing field, and its is good to see research taking this approach. The benefit of this approach is that it highlights the need for earlier intervention. The authors used well-validated measures (BDI-II for depression; IQCODE-SR for cognitive difficulties; General Activities Questionnaire for cognitive reserve; Global Activity Questionnaire for physical activity). The authors also compared CR preceding and during the COVID-19 pandemic and found no difference. The authors used the Center for Research in Language International Picture-Naming Project task for picture naming. The authors counterbalanced the presentation of the control tasks and language production tasks. Appropriate software and packages (R: tidyverse, lme4, emmeans) were used for analyses and the regression equations are described in enough detail. The authors pre-registered their study hypotheses, design, and statistical analyses on aspredicted.org. Their code is available on the GitHub repository.

There are some major issues with the study design, including a lack of description of how cognitive reserve was calculated, the use of novel online versions of cognitive tasks without existing norms, the lack of a representative sample, and most importantly, the confusion between brain and cognitive reserve. A Major Revision is recommended. It is requested that the authors address each of the comments below in their reply.

2.1. Major comments:

1. Introduction: The authors discuss brain networks and recruitment of additional brain networks, as ‘cognitive reserve’. This is controversial as there are separate constructs for ‘cognitive reserve’ and ‘brain reserve’. Can the authors please discuss the distinction between these concepts?

2. Method: The power calculation could be described more clearly. How many people were required to achieve 80% power and .05 alpha at a large effect size? What is the level required for a large effect size?

3. Method- Participants: Given how common anxiety & depression are throughout life, it is surprising that no participants reported history of or current psychiatric disorders. Were participants asked about severe conditions like schizophrenia and bipolar or more common conditions such as anxiety or mood disorders?

4. Method – Participants: The sample is not described in detail. What were their demographic details, such as ethnicity, and also education and time spent doing leisure activities (the variables used to calculate CR)?

5. Method- Participants: Why were only monolingual British English speakers chosen? The unrepresentativeness of the sample needs to be explicitly acknowledged in the Abstract and the Discussion.

6. Method: Please also provide descriptive statistics for the various measures used for the sample, as well as the internal consistency of the measures in the current sample.

7. Method: How was CR calculated? Were certain activities given more weight in the calculation? The method is vaguely described in the paper, but could be clearer.

8. Method: Where does the information about before and during COVID come from? Were the participants asked questions at multiple time points or was the questionnaire structured to ask about before and during COVID?

9. Method: The tasks used for semantic and action Verbal Fluency and the letters for Letter Fluency (S, M, P instead of F, A, S) appear to be novel. Was there any prior validation study conducted on the appropriateness of these materials?

10. Method: Inhibitory control, working memory, and cognitive processing speed tasks were adapted for an online format and normed within each age group. There were, however, only 30 people in each age category. This is not a representative sample for creating age group norms. Is there any prior research investigating the reliability and validity of the online versions of these tests? Were scores adjusted for sex or level of education?

11. Statistical analysis: Please provide a reference for the “Nelder Mead” optimiser method.

12. Results: Why was depression measured (BDI-II) and then not mentioned again in the manuscript? Did the authors control for depressive symptoms in their analyses?

13. Discussion: The authors state that middle-aged older adults might draw on CR for word-finding, but older adults might use other cognitive processes. Can you please explain what these processes might be?

14. Discussion: The authors mention interventions to promote CR in later life. Can you please expand on what are some promising interventions to do so?

2.2. Minor comments:

• When reported standard deviations, please use “SD = value” instead of “SD value” in the Results

• Please italicise statistical symbols (β, SD)

• Discussion, line 467-468: The grammar is odd in the first sentence of this paragraph. Can you please rephrase this?

6. PLOS authors have the option to publish the peer review history of their article (what does this mean?). If published, this will include your full peer review and any attached files.

Reviewer #1: **Yes: **Vitoria Piai

Reviewer #2: No

---

## [Author Response · Author response to Decision Letter 0]

19 Dec 2022

Please, find the rebuttal file attached as "Response to Reviewers".

---

## [Editor Report · Decision Letter 1]

3 Jan 2023

Getting the brain into gear: An online study investigating cognitive reserve and word-finding abilities in healthy ageing

PONE-D-22-23466R1

Dear Dr. Oosterhuis,

We’re pleased to inform you that your manuscript has been judged scientifically suitable for publication and will be formally accepted for publication once it meets all outstanding technical requirements.

Kind regards,

Katya Numbers, PhD, M.S., B.S.

Academic Editor

PLOS ONE

Additional Editor Comments (optional):

The authors have done a commendable job of incorporating the Reviewers' suggestions throughout the paper both in terms of larger conceptual issues and minor in-text and in-table revisions. It was my opinion that this was a strong paper originally; however, the discussion around nuanced differences between CR and brain reserve strengthens the paper and inclusion of some additional methodological information and limitations was important. I am pleased with how the authors have addressed the Reviewers' comments and suggestions and recommend the paper be accepted for publication. I congratulate the authors.
---

## [Editor Report · Acceptance letter]

1 Feb 2023

PONE-D-22-23466R1 

Getting the brain into gear: An online study investigating cognitive reserve and word-finding abilities in healthy ageing 

Dear Dr. Oosterhuis:

I'm pleased to inform you that your manuscript has been deemed suitable for publication in PLOS ONE. Congratulations! Your manuscript is now with our production department. 

Kind regards, 

on behalf of

Dr. Katya Numbers 

Academic Editor

PLOS ONE